# On the potential of vehicle-to-grid and second-life batteries to provide energy and material security

Fernando Aguilar Lopez [1] ✉, Dirk Lauinger [2,3], François Vuille[2] &
Daniel B. Müller [1]

The global energy transition relies increasingly on lithium-ion batteries for electric transportation and renewable energy integration. Given the highly concentrated supply chain of battery materials, importing regions have a strategic imperative to reduce their reliance on battery material imports through, *e.g.*, battery recycling or reuse. We investigate the potential of vehicle-to-grid and second-life batteries to reduce resource use by displacing new stationary batteries dedicated to grid storage. Based on dynamic material flow analysis, we show that equipping around 50% of electric vehicles with vehicle-to-grid or reusing 40% of electric vehicle batteries for second life each have the potential to fully cover the European Union's need for stationary storage by 2040. This could reduce total primary material demand from 2020–2050 by up to 7.5% and 1.5%, respectively, which could ease geopolitical risks and increase the European Union's energy and material security. Any surplus capacity could be used as a strategic reserve to increase resilience in the face of emergencies such as blackouts or adverse geo-political events.

Human societies require reliable access to energy at an affordable price to meet everyone's basic needs and comfort while enabling value-generating economic activities. The International Energy Agency (IEA) defines such access as energy security[1]. While the IEA was founded in the wake of the first oil crisis, today the global energy system is undergoing a structural transformation as society seeks to decarbonize to mitigate climate change and environmental degradation resulting from fossil fuels. This results in a new energy-material nexus in which the energy security of a region increasingly depends on its ability to reliably source enough materials to build clean energy infrastructure. In analogy with energy security, we term this reliable access to materials *"material security"*.

Renewable energy and electric vehicle technologies are essential to decarbonizing both the energy and transportation sectors. In Europe, most additional renewable electricity generation is expected to come from wind and solar since its geography limits the potential of other sources, such as hydropower and geothermal energy. A wide deployment of renewable electricity generation and electric transportation thus requires sufficient storage to (1) balance the intermittent production of wind and solar energy with electricity demand and (2) power the electric vehicles[2]. Within storage technologies, the industry is expected to largely remain committed to lithium-ion batteries (LIBs) for the foreseeable future because of their technological maturity and rapid cost decrease[3].

As societies shift from fossil fuels to LIBs for energy storage, energy security is increasingly predicated on a secure supply of LIB minerals such as lithium, nickel, and cobalt[4]. Concerns about material security thus become more pressing, as the LIB supply chain is dominated by just a few countries, even more so than the supply of fossil fuels[5]. China, for example, controlled over half of global lithium and cobalt processing in 2019[6]. The highly concentrated material supply, the ongoing COVID-19 pandemic, and geopolitical tensions such as the war in Ukraine led to significant price volatility in 2022: lithium prices roughly tripled compared to 2021, and nickel prices more than

[1]Norwegian University of Science and Technology, Trondheim, Norway. [2]Ecole polytechnique fédérale de Lausanne, Lausanne, Switzerland. [3]Massachusetts Institute of Technology, Cambridge, USA. ✉e-mail: fernando@siempreenergy.com

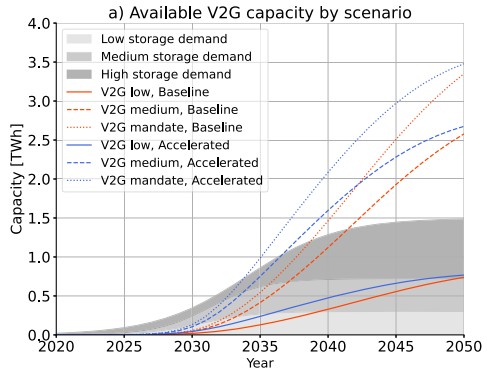
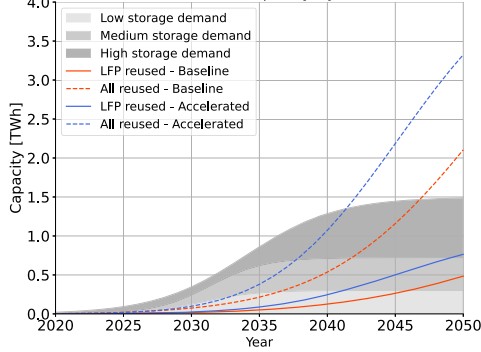

**Fig. 1 | Available energy storage capacity from V2G and SLB.** Potential capacity that can be offered by (**a**) V2G and (**b**) SLBs under a baseline and an accelerated EV penetration scenario. The EV penetration scenarios were defined according to scenarios proposed by the European Commission, ENTSOE, and the IEA (see SI 1.4). The expected demand for short-term stationary storage is based on scenarios by the ENTSOE and the European Commission (see SI 1.9).

doubled on a single day[7]. The increase in raw material prices led battery prices to increase for the first time since 2013[8]. The projected rapid growth of LIB demand in the coming decades puts further pressure on material supply and will require increased mining and processing of LIB minerals. This not only poses environmental and geopolitical risks but may simply not be feasible given the long lead times for the scale-up of production, refining, and recycling capacities[9]. Reducing the need for primary LIB materials is thus highly relevant for the energy transition.

Multifunctional use of EV batteries as storage for the electricity grid, either when the batteries are still in the EVs (vehicle-to-grid) or by reusing them after they are retired from the cars (second-life batteries) may reduce the need for additional stationary batteries. Vehicle-to-grid (V2G) can provide short-term storage when EVs sit idle, which is the case for over 90% of the time for privately owned cars[10]. The technical feasibility of V2G has been demonstrated in over 100 pilot projects since 2002[11,12]. Second-life batteries (SLBs) are EV batteries whose capacity has degraded to an extent, typically between 60% and 80% of the original capacity, making them unsuitable for continued use in EVs, but still serviceable as stationary storage for the grid[13,14]. The coupling of the transport and energy sector through V2G and SLBs holds the promise of providing more storage with fewer primary materials compared to using new batteries for grid support.

Xu et al. (2023) have concluded that electric vehicle batteries can satisfy stationary battery storage demand in the EU by as early as 2030, but they did not consider the resource implications of displacing new stationary batteries (NSBs) by V2G and SLBs[15]. Other studies have assessed the recycling potential of EV batteries: Kastanaki and Giannis (2023) found that SLBs in Germany and France could cover 27–70% of the stationary storage needs for photovoltaic systems and that recycled lithium could meet 5.2–6.2% of the lithium demand for EV batteries from the EU by 2030[16]. Similarly, Shafique et al. (2022) estimated that recycled lithium could meet 4-6% and 7-8% of the lithium demand from EV battery production by 2030 in China and the US, respectively[17]. In a global study, Xu et al. (2020) found that recycled lithium could reduce the cumulative lithium demand from EV battery production by 20–23% until 2050[18]. All these studies focused on the LIB demand for EV batteries, but their scope excluded the LIB demand for stationary storage. They thus could not assess how V2G or reuse would affect the primary material demand from both electric mobility and stationary storage. Studies on reuse have arrived at conflicting results. While Dunn et al. (2021) have claimed that reuse reduces the circularity of battery materials because it delays their availability for recycling[19], Bobba et al. (2020) suggested that reuse is an important circular economy strategy that maximizes the use of existing materials[20]. Aguilar Lopez et al. (2022) found that used LIBs from EVs can displace some of the LIBs that would otherwise be needed to cover the demand

for battery replacements[21]. However, these studies have not considered that SLBs can displace NSBs in providing stationary storage. Without these dynamics, the resource implications of implementing SLBs and V2G cannot be fully grasped. Since energy security is predicated on attaining material security, new modeling approaches are needed.

Based on dynamic material flow analysis[22,23], we develop a model that simulates the competition between NSBs, V2G, and SLBs in supplying a finite demand for electricity storage. Previous models have implicitly assumed an infinite demand for V2G and SLBs, which limited their ability to assess the resource implications of reuse and recycling. Using our model, we show that the material savings of substituting new batteries by SLBs more than compensate for the delay in recycling under current battery material recovery rates. Battery reuse thus reduces primary material demand even though it reduces the recycled content in new batteries. Overall, we find that V2G and SLBs could cover the demand for new stationary battery storage starting from 2035 and 2040 onward, which would reduce the total primary battery material demand from 2020 to 2050 by 7.5% and 1.5%, respectively. Without considering the dynamics of batteries needed for grid storage, the conclusions in the literature regarding reuse and recycling are based on an incomplete analysis that may lead to misinformed policies.

## Results

### Analysis of potential capacity: V2G and SLBs can each cover the expected needs for stationary battery storage

Figure 1 shows that in the long term V2G and SLBs each have the potential to exceed the demand for stationary battery storage for grid services by over a factor of two for even the highest demand scenario of the European Network of Transmission System Operators for Electricity (ENTSOE)[24]. In the early years, our model shows that NSBs are needed in all scenarios to meet the demand for grid storage. To meet the demand for grid storage in 2050, it would suffice to have about 40% of the EV fleet equipped with V2G, if 50% of all V2G-ready cars are plugged in at any given time and 50% of their battery capacity is made available for grid services (see SI 1.6), or to reuse about 45% of all end-of-life EV batteries. For comparison, about 60% of all end-of-life EV batteries are currently being reused in Norway[25]. From a volumetric point of view, V2G and SLBs may thus compete in the stationary storage market. The outcome of the competition will depend on their respective performance points, economics, and crucially, the timing of technology adoption. V2G has the potential to be deployed faster than SLBs, as EV batteries can already provide storage in their first "automotive" life, whereas SLBs only make use of EV batteries that have reached the end of their automotive life. However, regulatory impediments and societal resistance, *e.g.*, due to concerns about battery

degradation or range anxiety, could delay the adoption of V2G, which may allow stationary batteries to enter the market first.

In practice, the installed cumulative storage capacity of V2G, SLBs, and NSBs should not exceed the demand for stationary storage, as there may be no market for excess capacity. We thus developed a dynamic model, which is informed by the need for stationary storage on the demand side and by the availability of V2G and SLB capacity on the supply side. If demand exceeds supply, the model installs all available V2G and SLB capacity and fills the remaining demand with NSBs. If supply exceeds demand, the model first installs V2G and then SLBs until the demand is met. We prioritize V2G over SLBs because of its (i) potentially lower cost since V2G eliminates the need for batteries specifically dedicated to grid storage, (ii) earlier availability, and (iii) potentially lower needs for infrastructure updates since bidirectional charging capabilities can be embedded within EV drivetrains[11]. Any excess electric vehicles are not equipped with V2G, and any excess SLBs are directly collected for recycling. We assume that EVs that are equipped with V2G will be used for V2G throughout their automotive lives so that their owners do not forego profits from providing grid services. The methodology section provides a more formal description of the model.

## Material implications of different storage options: V2G reduces peak and cumulative primary material use more than SLBs
The total demand for battery materials will depend on the combination of V2G, SLBs, and NSBs used for grid storage. We first compare the yearly demand for battery materials from 2020–2050 of scenarios using exclusively NSBs, V2G, or SLBs (single technology scenarios). The goal is to estimate the maximum potential material savings of using V2G and SLBs compared to using NSBs. We thus consider a favorable context for each technology individually based on 1) a high storage demand scenario, 2) a V2G mandate scenario, and 3) the full reuse of EV batteries (see SI 1.3, 1.6, and 1.9 for a full description of the scenarios).

Figure 2 shows the installed storage capacity for each technology as well as the annual primary and recycled material demand from 2020–2050 under a baseline and an accelerated EV penetration scenario (see SI 1.4 for a full description of the scenarios). The lines show the ratio of newly installed storage each year compared to the available potential of V2G and SLBs individually. Initially, the deployment of V2G and SLBs is limited by the number of EVs in the overall vehicle fleet and by the low V2G adoption rate. The early demand for stationary storage is thus covered exclusively by NSBs in all scenarios. Although V2G and SLBs can fully cover the demand for new stationary storage in later years (2034 and 2038, respectively), some NSBs will still be in use in 2050 due to their long lifetime. In an accelerated EV penetration scenario, V2G and SLBs can cover the full demand for new storage capacity about 3 years earlier compared to the baseline. These results are in line with previous studies, which estimated that EV batteries could fully cover the demand for stationary storage starting in 2030 (Xu et al.)[15], and that SLBs could cover the stationary storage needed to support photovoltaic production in France and Germany from 2036 onward (Kastanaki and Giannis, 2023)[16].

We find that battery reuse reduces primary and peak primary material demand even though it reduces recycled content, i.e., the share of recycled materials in newly produced batteries, in the short term but reaches similar levels as the no reuse scenario once battery demand stabilizes. This finding contradicts previous studies, which claimed that reuse would increase primary material demand because they ignored the displacement of NSBs by SLBs; and reduced the recycled content of LIBs because they implicitly assumed an infinite demand for SLBs whereas we limit the amount of batteries that are reused by the demand for stationary storage[15,19,20,26]. However, the in-use time extension caused by reuse also reduces the

availability of recycled battery materials, which increases the primary material demand per new battery. Under current hydrometallurgical recycling, where some metals are recovered efficiently but aluminum, graphite, phosphorus, and lithium are mostly lost (see SI 1.8), the displacement of NSBs outweighs the higher primary material intensity of battery production (compare Fig. 2d with 2f and 2j with 2l). If there were no losses in the recycling process, then every old battery would result in a new battery. Since batteries lose some of their energy storage capability as they age (see SI 1.2 and 1.3), it would thus be more resource-efficient to recycle EV batteries directly after their automotive life to reduce primary material needs. It follows that the primary material savings from reuse decrease as battery recycling becomes more efficient. At direct recycling efficiencies (~90% efficiency for all materials, see SI 1.8), reuse is counter-productive purely from a resource use perspective (see supplementary Fig. 14). However, potential reductions in infrastructure needs, energy consumption, and greenhouse gas emissions may still justify reuse.

Assuming hydrometallurgical recycling, we find that V2G reduces the peak demand for primary materials compared to relying exclusively on NSBs by 5% under the baseline EV penetration scenario and by 10% under the accelerated EV penetration scenario. SLBs reduce the peak by about 1.5% in both EV penetration scenarios. Under the baseline EV penetration scenario, V2G and SLBs could fully cover the demand for new stationary battery storage starting from 2035 and 2040 onward, which would reduce the cumulative primary battery material demand from 2020 to 2050 by 7.5% and 1.5%, respectively. Exploring all scenario configurations and aggregating over all materials, we find that the cumulative material savings from recycling range between 9-18% (8-15% for Li) for hydrometallurgical recycling efficiencies and 17-32% (18-33% for Li) for direct recycling efficiencies. Xu et al. (2020) found this value to be 20-23% for lithium with a recycling efficiency of 95%[18], which is close to our direct recycling scenario. The higher range we observe can be attributed to several factors, including a faster electrification of the (European) vehicle fleet compared to their global study, making more batteries available for recycling sooner; and a slower growth rate of the vehicle fleet, which reduces the gap between the volume of batteries that become available for recycling and the demand for new batteries. Our lower-end estimations are due to the consideration of less efficient recycling technologies. Some materials, e.g., phosphorus, are recovered at low efficiencies even in direct recycling scenarios.

Battery reuse reduces the recycled content, i.e., the share of recycled materials from battery scrap in new batteries, during the growth phase in storage demand between 2020 and 2040. Regardless of battery reuse, the recycled content ranges from 25% to 45% by 2050, depending on the scenarios considered for EV and V2G adoption (see SI 3 for a breakdown per battery material). For lithium specifically, the recycled content ranges from 0.6-5% for hydrometallurgical recycling and 1–10% for direct recycling. This value overlaps with previous findings of 5.2-6.2% by Kastanaki and Giannis (2023)[16]. We attribute our wider range to the larger solution space we explored by including more parameters, such as stationary batteries and vehicle-to-grid.

## Competition between vehicle-to-grid and second-life batteries
We explore more closely the potential for SLBs and V2G to compete or complement each other in providing stationary storage by analyzing more conservative scenarios of battery reuse (only LFP chemistries are reused) and V2G adoption. Figure 3b, h show that low adoption of V2G (10% of vehicle sales by 2030 and 20% by 2040) can significantly reduce the need for new batteries while reducing the demand for SLBs by about half in 2050. A medium V2G adoption (25% by 2030 and 70% by 2040) almost eliminates the need for second-life batteries and penetrates the stationary storage market to a similar extent as the mandate scenario.

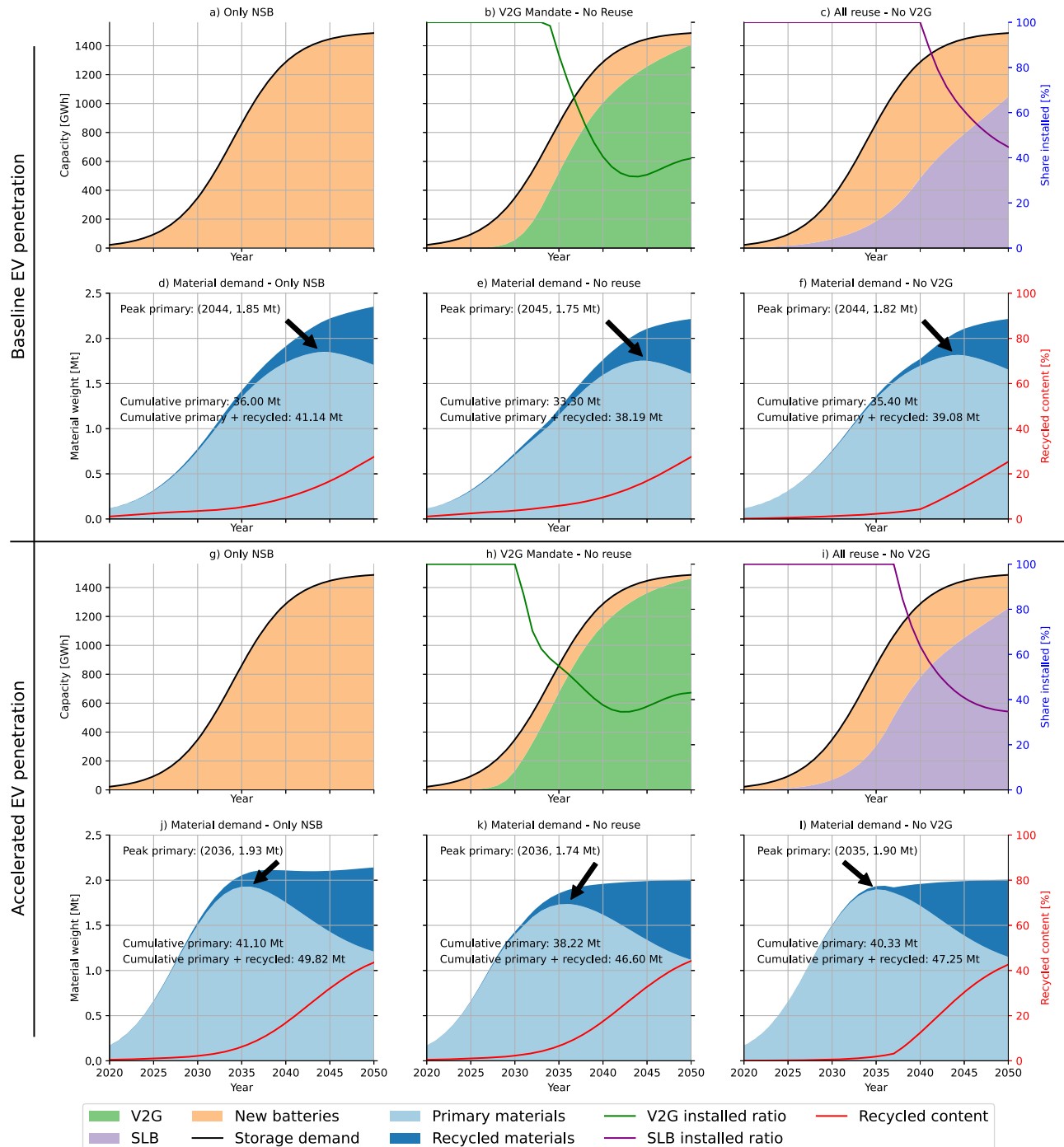

**Fig. 2 | Resource use per technology used to meet storage demand - High demand scenario.** Primary and recycled material use without V2G and SLB (**a**, **d** and **g**, **j**), with the V2G mandate only (**b**, **e** and **h**, **k**), and with reuse (**c**, **f** and **i**, **l**) of all battery chemistries only under the high storage demand scenarioMaterial demand includes batteries for EVs and for stationary storage. Share installed reflects the share of potentially available capacity from V2G and SLBs that was installed in any given year. The recycled content is the share of recycled materials from battery scrap in new batteries. Due to their long lifetime, NSBs that are installed in early periods to meet the demand for stationary storage may still be present in the stock in later periods when V2G and SLBs would have enough potential to cover the full storage demand. The six top figures show the baseline EV penetration scenario, and the bottom half shows the accelerated EV penetration scenario.

## Discussion

The large amount of battery storage needed for electric vehicles and for the electricity grid will increase the demand for battery materials. We showed that V2G and SLBs each have the potential to exceed the demand for stationary storage for grid support from about 2034 and 2041 onward, respectively. This remains valid after the European Network of Transmission System Operators for Electricity (ENTSOE) increased its projected energy storage requirements in response to the war in Ukraine[24], which is reflected in our high storage demand scenario. We find that, by reducing the need for new stationary batteries,

**Table 1 | Qualitative assessment of the advantages and risks related to each technology for providing grid services**

| Characteristic | V2G | SLB | NSB |
|---|---|---|---|
| Costs | **Advantage:** Bidirectional charger but no additional battery needed; potentially high aggregation and transaction costs. | **Neutral:** Lower investment costs than NSBs because batteries are reused; potentially high maintenance costs. | **Disadvantage:** Possibly more expensive than other options because EV batteries may have better economies of scale (~10x more batteries are needed for transportation than grid storage in the EU); could rely on less expensive chemistries. |
| Maintenance | **Advantage:** Already included in vehicles. | **Disadvantage:** State of health may vary and has higher uncertainty; may need frequent check-ups. | **Neutral:** More certain state of health; needs regular check-ups. |
| Social aspects | **Disadvantage:** Requires behavioral change; perceived fear of increased battery degradation; perceived loss of freedom in-vehicle use. | **Neutral:** If ownership is with the consumer, reuse can be desirable; perceived fire hazard. | **Advantage:** No social barriers identified. |
| Technical aspects | **Neutral:** Requires bidirectional charging infrastructure either built-in to the vehicle or external. | **Neutral:** Battery management system encryption may hinder the operation of repurposed batteries. | **Advantage:** A purpose-built battery management system may be an advantage. |
| Space requirements | **Advantage:** Culturally invisible because already embedded into EVs. | **Disadvantage:** Older technologies and a degraded state of health mean more space is needed than for NSBs. | **Neutral:** Requires some space for installation. |
| Material security | **Advantage:** Lowest primary and total material requirements. | **Neutral:** Reduces total and primary material needs at current recycling efficiencies but may be less beneficial as recycling technologies improve. | **Disadvantage:** Highest total and primary material requirements at current recycling efficiencies. |
| Energy security | **Advantage:** May exceed demand for grid storage by a factor of 2 by 2045 and could serve as energy reserve; stochastic availability means large numbers of cars are needed for reliable supply; can help enable large-scale renewable integration. | **Advantage:** May exceed demand for grid storage by a factor of 2 by 2048 and could be used to electrify other sectors; stationary and possibly more centralized than V2G; can help enable large-scale renewables. | **Advantage:** Potentially first mover; stationary and potentially more centralized than V2G; can help enable large-scale renewables. |
| Policy and regulations | **Disadvantage:** Market reforms are needed to allow for the participation of aggregators that can manage a large pool of individual vehicles and decentralized storage. | **Disadvantage:** Market reforms may be needed to allow for the participation of smaller players; competition with recyclers especially under recycled content regulations. | **Neutral:** This may require market reforms to allow for the participation of smaller players. |

The colors indicate whether a specific technology has an advantage, a neutral position, or a disadvantage on a particular point.

V2G can reduce the demand for primary battery materials by up to 7.5%. This is significant both from an economic and a geopolitical point of view, given the EU's heavy dependence on battery imports from Asia[9]. Beyond increasing material supply, V2G also contributes to energy security because the storage it provides helps the widespread integration of intermittent renewable energy, which reduces Europe's reliance on fossil fuel imports.

While a V2G mandate may enable the more timely and widespread adoption of V2G, we find that equipping 40% of all EVs with V2G would suffice to satisfy the expected needs of the electricity grid by 2050 if the owners of EVs with V2G make 50% of their battery capacity available to the grid and connect to the grid 50% of the time on average. This would still require significant political effort, albeit less than a mandate, since any large-scale V2G adoption requires significant changes in energy market regulations as well as in social acceptance and behavior. The latter barrier can potentially be reduced through the introduction of other technologies that enable the multifunctional use of batteries in a more tangible way to end users. Vehicle-to-home (V2H), for example, requires neither aggregators nor major changes in existing regulations. This technology allows photovoltaic prosumers to optimize the use of their solar energy by storing a potential surplus in their vehicles and discharging it later for self-consumption[27]. In blackout situations, V2H could power homes with islanding capabilities for several days[28], since the average residential electricity consumption in the EU is around 4.25kWh per person per day[29] and even small vehicles have battery capacities of at least 30kWh (see Supplementary Fig. 7). V2H could thus become an enabler for V2G as individuals would already be in possession of bidirectional chargers and accustomed to multipurpose their battery, which may render making batteries available to the grid a smaller leap to take compared to owners used to conventional charging.

Should a mandate also be considered, the abundant storage capacity could have strategic advantages: the vehicle fleet could act as an energy reserve in case of extreme events such as acute energy shortages. Currently, countries like Switzerland deal with such situations by purchasing mobile gas turbines and mandating hydropower reserves that can provide power for a limited number of weeks[30]. While the vehicle fleet will most likely not be able to provide electricity for such extended periods of time, it can still help to smoothen the transition from normal grid operations to emergency generators. High electricity prices during electricity shortages can be an incentive for vehicle owners to sell power from their vehicle batteries when it is needed most, provided they maintain a minimum state of charge for their driving needs. This would reduce the need for peak generation and, if there is not enough generation to meet demand, the need for load shedding, *i.e.*, the reduction of industrial activities to save energy, and hence help to prevent blackouts.

Considering the potential excess capacity of V2G and SLBs, the two technologies may compete for grid services both with each other and with NSBs. Table 1 compares the three technologies. The colors indicate whether a specific technology has an advantage (green), a neutral position (orange), or a disadvantage (red) on a particular point.

The EU is in the process of mandating recycling efficiencies, which correspond to current hydrometallurgical recycling, where graphite, phosphorus, manganese, and aluminum are not recovered, and only 35–70% of lithium is recovered[31]. By considering that SLBs are only installed if they displace NSBs, we challenge the prevalent belief that battery reuse increases primary material demand[19]. Under the EU recycling mandate, we find that the losses in the recycling process are substantial enough that reuse, followed by recycling after the end of reuse life, is a more resource-efficient strategy than recycling all EV batteries at the end of their automotive life to produce new batteries. If

the materials mentioned above were recovered efficiently, *e.g.*, with direct recycling (around 90% efficiency for all materials), the losses would become marginal, and reuse would become less attractive compared to using the recovered materials to produce (new, non-degraded) NSBs for the grid when solely considering resource efficiency (see Fig. 14 in the SI). However, battery recycling and production will still require additional infrastructure and energy consumption, and reducing those needs may still justify reuse. Further research is needed to investigate these aspects. Overall, given the expected recycling rates for the short- to mid-term, reuse will likely increase material security by reducing primary material demand.

Reuse has several other benefits not directly analyzed in this study. The creation of local markets for SLBs can help avoid end-of-life battery exports and increase the retainment of secondary battery materials in the EU. Beyond stationary storage for the electricity grid, any excess capacity of SLBs could serve to electrify sectors that cannot afford other forms of stationary storage, such as remote off-grid areas, and provide uninterrupted power supply to critical infrastructures such as hospitals and water distribution systems. Finally, by delaying the large-scale recycling of EV batteries, reuse, on the one hand, provides the industry with more time to increase both its recycling efficiency and capacity but, on the other hand, limits the amount of scrap material available for refining recycling processes.

The proposed EU regulation to mandate a minimum recycled content in batteries from 2026 onward[31] would create a disincentive to reuse batteries since large volumes of battery scrap will be needed to meet the recycled content mandate. As the European EV fleet has been and is set to continue growing rapidly, there is a risk that there will not be enough scrap available to meet the mandate until the EV stock stabilizes (see Figs. 2 and 3), especially if retired batteries are reused. This might present an incentive for shorter battery lifetimes to maximize scrap availability to meet the requirements. The proposed regulation in its current form could thus paradoxically increase primary material demand. On the positive side, a recycling mandate may provide certainty about future developments in the battery industry and spur investments in recycling. It also presents a first step towards material-specific recycling targets, which can be crucial to ensure the recycling of materials whose recovery may not currently be economical but on which the EU is highly import-dependent. We note, however, that the mandate mainly targets materials such as nickel, cobalt, and copper, which are already efficiently recovered, but does not require the recycling of materials such as manganese, phosphorus, aluminum, graphite, and silicon, whose recovery would allow for recycling to become even more resource efficient than reuse in the long term.

Overall, the risk of shortages of LIB material supply[26,32,33] could mean that the EU will not only need to decide on the pace of electrification for different industrial sectors but may also need to decide how much of a given material should be used for batteries and how much should be used for other applications. The case of phosphorus, which is needed for LFP batteries as well as agriculture fertilizers, may pose a particularly pertinent issue. The same can be said for nickel in the stainless-steel industry and for aluminum in its various applications, which include building and construction, transportation, and electric cables[34]. Poorer countries may suffer disproportionately more from higher resource prices, especially for essential commodities such as fertilizer. Any reduction in primary material use could thus contribute to important systemic risk mitigation on various supply chains beyond lithium-ion batteries.

Besides lithium-ion batteries, the EU plans to provide short-term flexibility to its electricity grids by modulating the operating points of electrolyzers[24,35]. The electrolyzers will thus have to be oversized to produce the same amount of hydrogen as if they were running at full capacity. Such electrolyzers commonly use platinum and iridium as catalysts. Iridium is mostly mined as a by-product of platinum. The supply chain of platinum is even more concentrated than the supply chain of battery materials: about 80% of the global platinum production comes from South Africa[36,37]. Since V2G and SLBs combined can provide up to four times the projected needs for battery storage by 2050, the excess storage potential could be used to provide some of the short-term flexibility that is expected to come from electrolyzers. This would allow electrolyzers to run more closely to full capacity and hence make more efficient use of platinum and iridium, which reduces the reliance on the concentrated platinum supply chain and increases mineral security. Other options include the use of alternative technologies: supercapacitors, compressed air-storage technologies, and fly-wheels could all be used to diversify the material use portfolio and, therefore, reduce the risk of supply constraints of specific materials.

Our study relies on assumptions for a range of parameters that influence the availability of energy storage and battery material needs. These parameters are affected by several social and technological factors. Autonomous driving, for instance, could considerably reduce the availability of EVs for grid services by increasing their driving time, thereby reducing time plugged into the grid. This effect could be exacerbated by a potential reduction of the overall vehicle fleet, which would result in fewer vehicles available for V2G and a lower need for EV batteries. In fact, reducing the size of the EV fleet seems like the most efficient way to reduce potential material supply bottlenecks, as EVs demand an order of magnitude more batteries than the grid. Reducing the weight of EVs could increase their energy efficiency and allow for the downsizing of battery capacity without comprising range, which would reduce battery material needs. However, this should be investigated further as the problem shifts to other metals such as aluminum for light weighing could occur[38]. Car bans in city centers, increased investment in public transportation and bike infrastructure, and smart city design could reduce both the vehicle fleet and overall energy consumption, which would contribute to increased energy and material security.

Throughout this study, we compared aggregate storage demand with aggregate storage availability without considering bottlenecks in the electricity grids that connect centers of storage demand with centers of storage supply. We thus overestimated the effective storage demand that V2G may supply. However, since V2G has the potential to supply more than twice the anticipated demand for stationary battery storage in the long term (see Fig. 1), it seems likely that V2G could fully supply the storage demand in the long term, even when accounting for bottlenecks. Future work could combine our material flow analysis with spatially explicit energy system models[2] that compute storage needs at various points throughout Europe.

Overall, our study showed the importance of considering the demand for both electric transportation and grid storage when assessing future resource needs for lithium-ion batteries. Securing a stable supply of these resources is a strategic concern for Europe. On the one hand, we found that policies that were designed to increase self-sufficiency, such as the proposed EU regulation on battery recycled content, may backfire because they disincentivize battery reuse and thus increase the demand for primary battery materials. On the other hand, considering the interplay of recycling along with multifunctional battery use technologies reveals opportunities to reduce total primary material needs and bolster both Europe's energy and material security.

## Methods
### System definition
We investigate the LIB system related to the passenger vehicle fleet and stationary energy storage in the European Union (including the European Free Trade Association) using a yearly resolution from 1950 to 2050 (Fig. 4).

The system differentiates three layers: vehicle, batteries, including their storage capacity, and battery materials. Vehicles are classified by

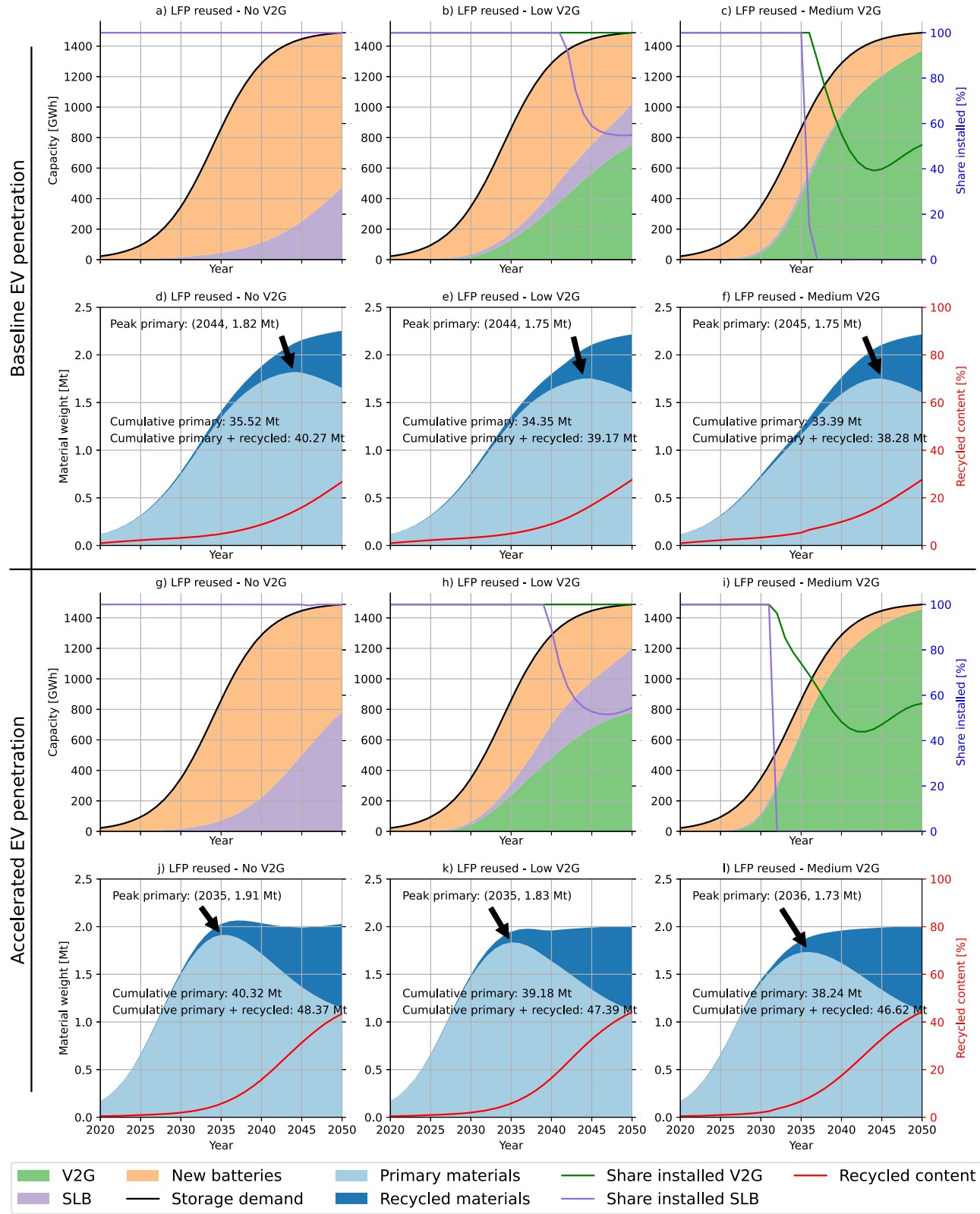

Resource use per technology used to meet storage demand - High demand scenario

their drive train and grouped into internal combustion engine vehicles (ICEV), battery electric vehicles (BEV), plug-in hybrid electric vehicles (PHEV), and other types of vehicles such as fuel cell vehicles. BEVs and PHEVs are further divided based on the size of their batteries. We distinguish whether BEVs are equipped with V2G technology. PHEVs are excluded from V2G because of their limited battery capacity. EV batteries are segmented by chemistry to estimate the raw materials needed to produce them.

**Fig. 3 | Resource use per technology to meet storage demand - High demand scenario.** Primary and recycled material used for the no (**a**, **d** and **g, j**), low (**b**, **e** and **h**, **k**), and medium V2G (**c**, **f** and **i, l**) scenarios when LFP chemistries are reused and the storage demand is high. V2G displaces the need for SLBs since it has priority in providing stationary storage. Material demand includes batteries for EVs and for stationary storage. Share installed reflects the share of potentially available capacity from V2G and SLBs that was installed in any given year. The recycled content is the share of recycled materials from battery scrap in new batteries. Due to their long lifetime, NSBs that are installed in early periods to meet the demand for stationary storage may still be present in the stock in later periods when V2G and SLBs would have enough potential to cover the full storage demand. The six top figures show the baseline EV penetration scenario, and the bottom half shows the accelerated EV penetration scenario.

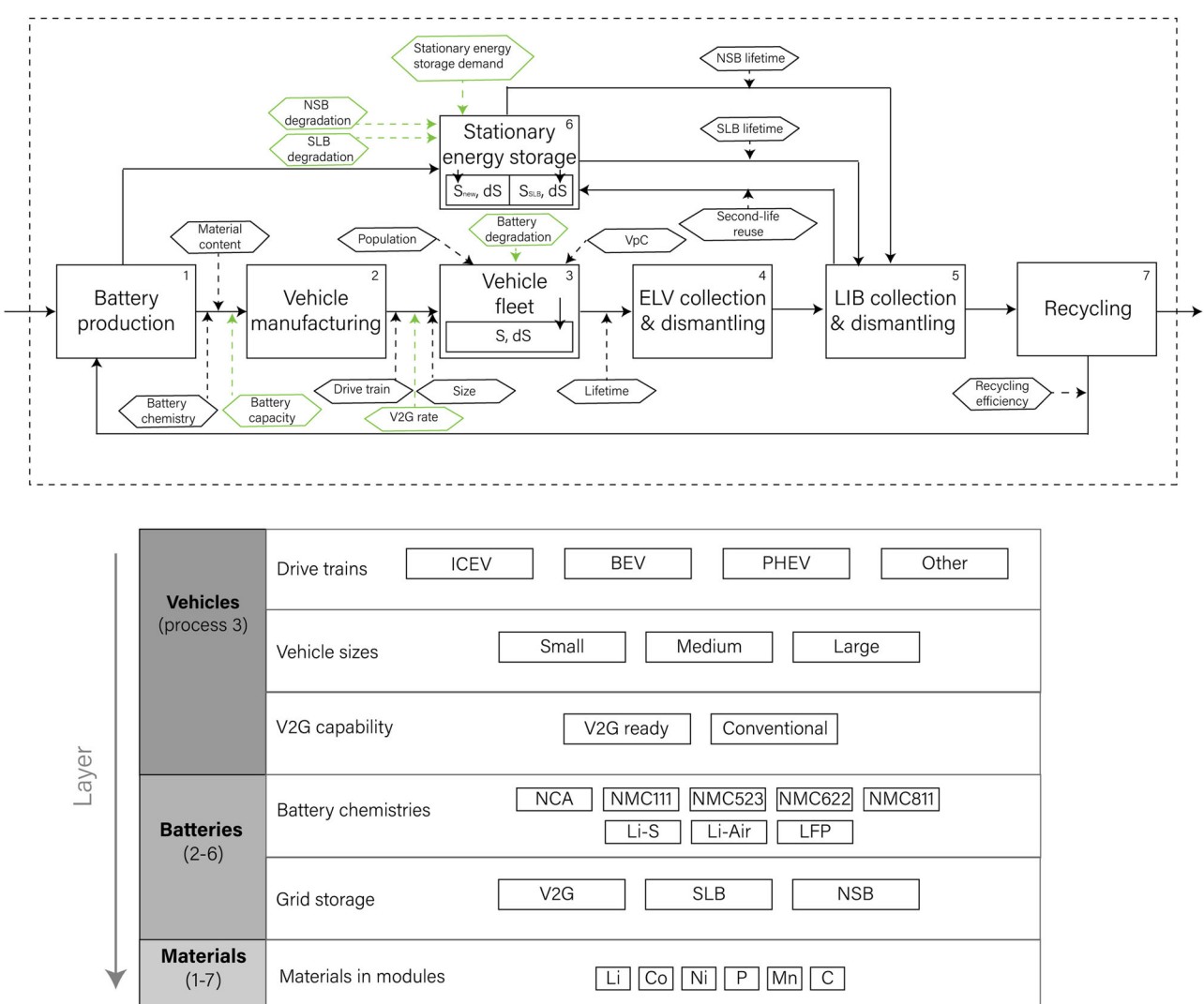

**Fig. 4 | System definition and parameters of the European vehicle and stationary storage infrastructure.** The top part shows the main processes in squared boxes and system parameters in hexagonal boxes. Energy and material parameters are marked in green and black, respectively. The bottom part shows the layers included in the model, which can be balanced for the processes marked in parenthesis.

Once electric vehicles reach their end-of-life (EOL), their batteries can be either reused for grid storage in a second life or go directly to recycling. Second-life batteries (SLBs) eventually reach EOL as well and are collected for recycling. In addition to SLBs, new stationary batteries (NSBs) are produced to cover the demand for grid storage. Since LFP chemistries enjoy a longer lifetime and higher safety than other common LIB chemistries, we consider that all NSBs are based on a LFP chemistry. Upon reaching EOL, all NSBs are collected for recycling.

### Model development and calibration

We rely on a stock-driven material flow analysis methodology[23] to quantify the dynamics of the vehicle fleet for alternative scenarios. By multiplying the population projections from the United Nations[39] with a self-defined baseline scenario for vehicle ownership based on historical data[40], we calculate the total vehicle stock in the EU (see SI 1.1).

We assume that the lifetime of all vehicles follows a normal distribution with a mean of 15 years and a standard deviation of 5 years, which corresponds to the current lifetime of ICEVs in the EU[41]. Initially, EVs were thought to have a shorter lifetime than ICEVs due to battery degradation but recent experience suggests these concerns were exaggerated[18,26,42,43]. We neglect the impact of vehicle-to-grid on battery lifetime since V2G tends to lead to more gentle charging of the battery and smarter management of its state of charge[44,45]. While some chemistries, such as LFP, are generally considered to be longer lasting, we neglect chemistry-related degradation effects to reduce the complexity of the model as well as exposure to the highly uncertain assumptions on the future battery chemistry mix. However, we

considered a scenario in which only LFP batteries are reused to reflect their better suitability for a second life. Finally, we assume that EV batteries have 80% of their initial capacity left when they reach the end of their automotive life[46,47] (see SI 1.2 for further explanations).

The penetration of BEVs and PHEVs into the vehicle fleet is calculated as a share of total vehicle sales and is based on projections by the International Energy Agency (IEA)[48], the European Commission (EC)[35], and the European Network of Transmission System Operators for Electricity (ENTSOE)[24,49]. In our baseline EV penetration scenario, BEVs will make up 47% of the total European vehicle sales by 2035. An accelerated EV penetration scenario assumes that 74% of the sales will consist of BEVs by 2035. In this scenario, BEVs and fuel-cell electric vehicles together account for 81% of vehicle sales by 2035, while PHEVs and ICEVs account for 19% of vehicle sales. The accelerated scenario is in line with the European Council's goal of a 100% CO2 emission reduction for new cars by 2035[50] if the internal combustion engines in the PHEVs and ICEVs are powered by carbon-neutral fuels (see SI 1.4). We consider that BEVs can have small (33 kWh), medium (66 kWh), or large (100 kWh) batteries, while PHEVs have smaller batteries of 8 kWh, 12 kWh, and 17 kWh, respectively (see SI 1.5).

Beyond a no-V2G baseline, we consider three scenarios for the share of BEV sales that are equipped with V2G technology. In all scenarios, the V2G penetration saturates around 2035 at levels of 20%, 70%, and 90% for a low, medium, and mandate scenario, respectively. We assume that all owners of vehicles equipped with V2G will deliver grid services. Specifically, we assume that 50% of all vehicles equipped with V2G are parked and connected to the grid at any given time and that the owners make 50% of the battery storage available for V2G (see SI 1.6 for a justification of these assumptions).

For the split of battery chemistries in new EVs, we follow a baseline scenario defined by Bloomberg New Energy Finance[51] (SI 1.7). When calculating the raw material needs for battery production, we aggregate over the battery materials (Li, Co, Ni, P, Mn, graphite) to reduce the sensitivity of our results to the individual materials contained in future battery chemistries. In the SI, we report the material needs for each element (SI 3), and the material content of each battery chemistry can be found as a separate supplementary table in Excel format in addition to the numpy arrays used in the numerical implementation of the model.

We examine three scenarios for the reuse rate of EV batteries: no reuse, reuse of LFP batteries only given their inexpensive materials and hence low value for recycling, and reuse of all batteries. All SLBs are assumed to remain in stationary applications until their storage capacity degrades to 60% of the initial storage capacity, which is assumed to happen within 6 years with a standard deviation of 2 years, in accordance with the scant literature on the topic[46] (SI 1.3). Once batteries are collected for recycling, they undergo a hydrometallurgical process with material-specific recycling efficiencies (see SI 1.8).

Assumptions about the demand for battery storage for the electricity grid are based on reports by the EC[35,52,53] and by the ENTSOE[24,49]. We consider a low scenario reaching 0.3TWh by 2040, a medium scenario reaching 0.7TWh by 2040, and a high scenario reaching 1.5TWh (see SI 1.9).

The choice between stationary storage technologies (V2G, SLBs, and NSBs) works in two different ways: (1) using an unconstrained model according to traditional material flow analysis methodology[22,54] for the volumetric analysis in Fig. 1, and (2) using a demand-constrained model for the analysis of the interlinkages between different technologies in Figs. 2 and 3.

In the unconstrained model, all available V2G and SLB capacity is installed in each year regardless of the demand for stationary storage. The model assesses the maximum storage potential from V2G and SLBs; no NSBs are installed in this approach. The maximum potential is compared with the demand for stationary storage *ex post*.

In the demand-constrained model, the demand for stationary storage must always be met through a combination of V2G, SLBs, and NSBs. The capacity installed in each year is decided based on a hierarchical model that gives preference to (1) equipping new cars with V2G and (2) reusing EOL EV batteries as SLBs. Any surplus demand is covered with NSBs and any surplus in V2G or SLB availability is not installed but recycled directly. The installed capacity of each storage type is adjusted for battery degradation and for battery outflows in each year. The transfer coefficients in the unconstrained model are fixed *ex ante* and adjusted dynamically in the demand-constrained model based on the available supply of V2G and SLBs. See SI 1.10 for a more formal description.

All recycled materials are used for manufacturing new batteries. As we are interested in the maximum amount of recycled battery materials that the EU may expect, we assume perfect collection and no trade of spent batteries.

## Reporting summary
Further information on research design is available in the Nature Portfolio Reporting Summary linked to this article.

## Data availability
All data generated in this study have been deposited in the Zenodo database under the accession code https://doi.org/10.5281/zenodo.10732563. All data are available under the Creative Commons Attribution 4.0 International license.

## Code availability
The model was implemented in Python 3 and relies on the *Open Dynamic Material Systems Model* framework[55]. All code is available under the Creative Commons Attribution 4.0 International license in the following repository https://doi.org/10.5281/zenodo.10732563

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

## Acknowledgements

This work was partly funded by the Institut Vedecom (F.V.), the Norwegian Research Council under the BATMAN project 299334 (D.B.M.), and Peugeot (F.V.). We thank Yifei Liu for editing the final manuscript. We acknowledge fruitful discussions with Damien Pierre Sainflou and François Colet.

## Author contributions

All authors were involved in the design of the study; D.L. and F.V. conceived the study, F.A.L. and D.L. collected the data, F.A.L. developed the model and generated the figures; F.A.L. and D.L. developed the scenarios; F.A.L. and D.L. led the writing with assistance from F.V. and D.B.M.; all authors interpreted the results, reviewed the paper, and approved the final version of the manuscript.

## Funding

## Competing interests

The authors declare no competing interest.

## Inclusion & Ethics statement

Authorship of this paper has been granted according to Nature Portfolio standards.
