## [Peer Review File · Nature Communications]

REVIEWER COMMENTS

Reviewer #1 (Remarks to the Author):

The paper tries to provide an analysis on dynamic material flow that vehicle-to-grid and second-life batteries both have the potential to fully cover the EU's need for stationary storage. Although the paper is relatively interesting, the paper requires further significant improvement and correction.

1. The novelty of the review paper needs to be justified and clearly defined. It includes the clear difference with the available literature and previous review works. The authors are asked to provide the limitations of the previous correlated review works and then link those limitations to the current ideas and contributions of the current work.

2. In the abstract, please also include the main or representative quantitative results, not only qualitative explanation. Therefore, the readers can understand accurately the contents of the work. In addition, the current abstract mainly focuses on the background of the review. You need to focus much more on the main contents and findings of the review works. In addition, a brief discussion on the main findings is also required.

3. Several references are lumped together without providing sufficient description for each of them. It gives nothing to the readers. Please provide a short description or descriptor for each used reference, hence, the readers can understand the content of each used reference.

4. You need to extend the contents and discussion of your work to be much more concise and comprehensive. Some discussions regarding dynamic EV adoption potential, user behaviors for EV purchase and V2G participation, secondary battery market, and possible/applicable technologies for V2G and secondary battery controls must be provided in the context of the European case.

5. In the results and discussion, please also provide a comprehensive comparison with the results from other studies and works. Therefore, the visibility of the analysis results can be measured objectively.

6. In the context of V2G, fluctuation from renewable energy becomes a very important influence to the system. The interesting point is that, the fluctuation conditions in each country in the European cases are significantly different due to the ratio of renewable energy to demand, grid scale, and etc. You need also to discuss this issue to measure the applicability of V2G in different countries in EU.

Reviewer #2 (Remarks to the Author):

Reviewer #1: The authors present a distribution delay forecasting model to forecast several characteristics of waste batteries from full-electric vehicles (BEV) in the EU for the coming ten years. The lifetime distribution is similar to that used in <https://doi.org/10.1038/s43246-020-00095-x>, hence clarifications on the innovative contribution beyond the latter publication have to be highlighted better, taking into account that materials are sourced in a world market. Furthermore, it is not clear why the Economic burden of recycling LIBs was excluded from the scope of the study, even though they use the same cell chemistry in BEV batteries. The forecasting model predicts the waste stream size, second use potential, recycling potential and the potential of vehicle-to-grid. Results from this study confirm those from previous studies for different regions and time horizons.

General comments:

- Lines 37-43 describe previous work. This should include some of the recent literature on EOL i.e. <https://doi.org/10.1016/j.jclepro.2023.136349>; <https://doi.org/10.1016/j.resconrec.2021.106061> etc. in the

Introduction section:

- The Introduction part should be revised to improve its logical structure.

- There is not enough background regarding why models/functions for estimating EOL LIBs are important, what has been done and found in the previous studies, and what is still lacking.

- The authors presented some previous studies from line 58, but these contents seem too separate from the above contents. No good transitions and clear logic in these parts.

- Additionally, what is the novelty of this study to differentiate it from the previous research? Therefore, the current Introduction doesn't provide clear and comprehensive descriptions of the necessity and importance of this study

The Methodology part:

- Line 339: We assume that the lifetime of all vehicles follows a normal distribution with a mean of 15 years; however in reality some of the batteries have issues in their first 5 years and reach their EOL. why the authors neglected it. There are no descriptions of this method/model, so please clarify this.

- Line 368, what is C in the battery materials?

- In the methodology section, 1.8 Recycling efficiencies why C is not mentioned, as C is mentioned for battery materials?

- As there were different types of materials were used in each LIBs, however, there is no clear description of materials composition of materials for batteries in this section, this should be elaborated.

- How it was assumed for vehicle-to-grid for each battery and vehicle type? it should be demonstrated.

The Results and discussion part:

- Fig.3 should be revised to make it better.

- The descriptions of the results should be made clear to indicate which scenario is the data/results for.

- The article considers the second use of lithium-ion batteries scrapped from electric vehicles and sets different utilization scenarios. What is the lifetime of the second use of a battery? How is it determined?

The whole manuscript should be checked very carefully since there are many inconsistent terms, error notations and expressions, and even a lack of important data sources. All of these make it difficult to read and weaken the quality of this paper

Reviewer #3 (Remarks to the Author):

Dear authors and editor,

I have reviewed the manuscript and supporting information drafts entitled “On the potential of vehicle-to-grid and second-life batteries to provide energy and material security”, and I find the research work to be of high quality, well-reported and transparent. The topic area is of high relevance, and the paper provides new knowledge, especially in terms of the material demand impacts of using vehicle batteries for vehicle-to-grid services. Other than a few minor corrections, my recommendation is that this submission should be accepted for publication.

Proposed minor revisions:

Main manuscript text row 59 of main manuscript: The abbreviation “NSB” has not yet been introduced, this is done later in the paragraph, on row 62. Please swap this between the first and second instance.

Main manuscript Figure 2: The red line representing the projected recycled content share is not included in the legend. Differing from the other line colors, the meaning of the red line can be figured out from the figure itself, but for readability and clarity, it is recommended to include the red line meaning in the legend.

Main manuscript Table 1, Row Costs, Column SLB: It is proposed to reformulate “used” to “reused”.

Main manuscript text row 236-238: It is claimed that a benefit of delaying large-scale recycling of EV batteries is that more time becomes available for improving recycling procedures. I question this argument and find it superficial. It can equally well be argued that learning typically relates to accumulated throughput, and without large volumes of batteries coming to recycling, there will be less

learning and less improvement, i.e., calendar time is of less importance. My recommendation to the authors is to revisit this specific point and revise the argument.

Main manuscript text row 236-238: It is recommended to remove a part of the last sentence in the paragraph, i.e. rephrase “which would contribute to a more sustainable future with increased energy and material security.” to “which would contribute to increased energy and material security.”

The suggested contribution to a “sustainable future” is vague and speculative, lowering the credibility of the remaining claim.

SI text row 73: The heading not connected to adjacent text. Please shift the heading to the following page.

Response letter

Reviewer #1: [your comments in italics]

The paper tries to provide an analysis on dynamic material flow that vehicle-to-grid and second-life batteries both have the potential to fully cover the EU's need for stationary storage. Although the paper is relatively interesting, the paper requires further significant improvement and correction.

1. The novelty of the review paper needs to be justified and clearly defined. It includes the clear difference with the available literature and previous review works. The authors are asked to provide the limitations of the previous correlated review works and then link those limitations to the current ideas and contributions of the current work.

Thank you for highlighting this important point. We clarified the gap that this study is addressing and added additional references, including those suggested by reviewer 2, to justify and clarify the novelty. The literature review paragraph now reads:

“Xu et al. (2023) have concluded that electric vehicle batteries can satisfy stationary battery storage demand in the EU by as early as 2030 but they did not consider the resource implications of displacing new stationary batteries (NSBs) by V2G and SLBs¹⁵. Other studies have assessed the recycling potential of EV batteries: Kastanaki and Giannis (2023) found that SLBs in Germany and France could cover 27-70% of the stationary storage needs for photovoltaic systems, and that recycled lithium could meet 5.2-6.2% of the lithium demand for EV batteries from the EU by 2030¹⁶. Similarly, Shafique et al. (2022) estimated that recycled lithium could meet 4-6% and 7-8% of the lithium demand from EV battery production by 2030 in China and the US, respectively¹⁷. In a global study, Xu et al. (2020) found that recycled lithium could reduce the cumulative lithium demand from EV battery production by 20-23% until 2050¹⁸. All these studies focused on the LIB demand for EV batteries, but their scope excluded the LIB demand for stationary storage. They thus could not assess how V2G or reuse would affect the primary material demand from both electric mobility and stationary storage. Studies on reuse have arrived at conflicting results. While Dunn et al. (2021) have claimed that reuse reduces the circularity of battery materials because it delays their availability for recycling¹⁹, Bobba et al. (2020) suggested that reuse is an important circular economy strategy that maximizes the use of existing materials²⁰. Aguilar Lopez et al. (2022) found that used LIBs from EVs can displace some of the LIBs that would otherwise be needed to cover the demand for battery replacements²¹. However, these studies have not considered that SLBs can displace NSBs in providing stationary storage. Without these dynamics, the resource implications of implementing SLBs and V2G cannot be fully grasped. Since energy security is predicated on attaining material security, new modelling approaches are needed.”

15. Xu, C. *et al.* Electric vehicle batteries alone could satisfy short-term grid storage demand by as early as 2030. *Nat Commun* **14**, 119 (2023).

16. Kastanaki, E. & Giannis, A. Dynamic estimation of end-of-life electric vehicle batteries in the EU-27 considering reuse, remanufacturing and recycling options. *Journal of Cleaner Production* **393**, 136349 (2023).
17. Shafique, M., Rafiq, M., Azam, A. & Luo, X. Material flow analysis for end-of-life lithium-ion batteries from battery electric vehicles in the USA and China. *Resources, Conservation and Recycling* **178**, 106061 (2022).
18. Xu, C. *et al.* Future material demand for automotive lithium-based batteries. *Commun Mater* **1**, 99 (2020).
19. Dunn, J., Slattery, M., Kendall, A., Ambrose, H. & Shen, S. Circularity of lithium-ion battery materials in electric vehicles. *Environ. Sci. Technol.* **55**, 5189–5198 (2021).
20. Bobba, S. *et al.* Bridging tools to better understand environmental performances and raw materials supply of traction batteries in the future EU fleet. *Energies* **13**, 2513 (2020).
21. Aguilar Lopez, F., Billy, R. G. & Müller, D. B. A product–component framework for modeling stock dynamics and its application for electric vehicles and lithium-ion batteries. *Journal of Industrial Ecology* **26**, 1605–1615 (2022).

2. *In the abstract, please also include the main or representative quantitative results, not only qualitative explanation. Therefore, the readers can understand accurately the contents of the work. In addition, the current abstract mainly focuses on the background of the review. You need to much more focus on the main contents and findings of the review works. In addition, a brief discussion on the main findings is also required.*

Thank you for these important suggestions. We have reduced the background information in favour of adding more quantitative details about our study's main findings. The abstract now reads as follows:

“The global energy transition relies increasingly on lithium-ion batteries for electric transportation and renewable energy integration. Given the highly concentrated supply chain of battery materials, importing regions have a strategic imperative to reduce their reliance on battery material imports through, e.g., battery recycling or reuse. We investigate the potential of vehicle-to-grid and second-life batteries to reduce resource use by displacing new stationary batteries dedicated to grid storage. Based on dynamic material flow analysis, we show that equipping around 50% of EVs with vehicle-to-grid or reusing 40% of EV batteries for second life each have the potential to fully cover the EU's need for stationary storage by 2040. This could reduce total primary material demand from 2020–2050 by up to 7.5% and 1.5%, respectively, which could ease geopolitical risks and increase the EU's energy and material security. Any surplus capacity could be used as a strategic reserve to increase resilience in the face of emergencies such as blackouts or adverse geo-political events.”

3. *Several references are lumped together without providing sufficient description for each of them. It gives nothing to the readers. Please provide a short description or descriptor for each used references, hence, the readers can understand the content of each used references.*

Thank you for your helpful advice. We now provide more detail on the references in the literature review, see the answer to your first point. In other places, we left several references lumped together, when we felt that a claim needed robust backing and that a detailed description of each reference individually would be unduly verbose.

4. *You need to extend the contents and discussion of your work to be much more concise and comprehensive. Some discussions regarding dynamic EV adoption potential, user behaviors for EV purchase and V2G participation, secondary battery market, and possible/applicable technologies for V2G and secondary battery controls must be provided in the context of the European case.*

Thank you for pointing this out. In the discussion and conclusions section, we added a paragraph about the impact of spatial aggregation on the effective capacity that V2G may deliver in Europe, which also addresses your last comment, and we now discuss vehicle-to-home as a potential first step toward vehicle-to-grid:

“While a V2G mandate may enable the more timely and widespread adoption of V2G, we find that equipping 40% of all EVs with V2G would suffice to satisfy the expected needs of the electricity grid by 2050, if the owners of EVs with V2G make 50% of their battery capacity available to the grid and connect to the grid 50% of the time on average. This would still require significant political effort, albeit less than a mandate, since any large-scale V2G adoption requires significant changes in energy market regulations as well as in social acceptance and behavior. The latter barrier can potentially be reduced through the introduction of other technologies that enable the multifunctional use of batteries in a more tangible way to end users. Vehicle-to-home (V2H), for example, requires neither aggregators nor major changes in existing regulations. This technology allows photovoltaic prosumers to optimize the use of their solar energy by storing a potential surplus in their vehicles and discharging it later for self-consumption²⁷. In blackout situations, V2H could power homes with islanding capabilities for several days²⁸, since the average residential electricity consumption in the EU is around 4.25kWh per person per day²⁹ and even small vehicles have battery capacities of at least 30kWh (see SI Figure 7). V2H could thus become an enabler for V2G as individuals would already be in possession of bidirectional chargers and accustomed to multipurposing their battery, which may render making batteries available to the grid a smaller leap to take compared to owners used to conventional charging.”

27. Liu, C., Chau, K. T., Wu, D. & Gao, S. Opportunities and Challenges of Vehicle-to-Home, Vehicle-to-Vehicle, and Vehicle-to-Grid Technologies. *Proc. IEEE* **101**, 2409–2427 (2013).

28. Pinto, J. G. *et al.* Bidirectional battery charger with Grid-to-Vehicle, Vehicle-to-Grid and Vehicle-to-Home technologies. in *IECON 2013 - 39th Annual Conference of the IEEE Industrial Electronics Society* 5934–5939 (IEEE, 2013).

29. Tsemekidi Tzeiranaki, S. *et al.* *Energy consumption and energy efficiency trends in the EU-28 for the period 2000-2016*. (Publications Office of the European Union, 2018).

The SI includes detailed information and discussions about electric vehicle penetration (SI 1.4), vehicle and battery sizes (SI 1.5), vehicle-to-grid participation (SI 1.6), battery chemistries (SI 1.7), and stationary storage demand (SI 1.9).

Finally, we compare V2G, SLBs, and NSBs qualitatively in Table 1 based on several criteria such as costs, space requirements, social acceptability, and need for policy reforms.

5. In the results and discussion, please also provide a comprehensive comparison with the results from other studies and works. Therefore, the visibility of the analysis results can be measured objectively.

Thank you for your suggestion. Overall, we arrive at similar results as previous studies in terms of (1) the storage potential from EV batteries, (2) recycled content, and (3) cumulative primary material savings from recycling. We generally report a wider range on these metrics than previous studies because we consider more parameter variations, *e.g.*, recycling efficiencies, vehicle-to-grid, and new stationary batteries. This suggests that the novelty in our approach, *i.e.*, considering the displacement of NSBs by V2G and SLBs, has only a limited impact on the three comparison metrics listed above. We find that electric vehicles will need about ten times more battery materials than grid storage, hence it is not surprising that the added demand for grid storage does not significantly change the results obtained when considering EV battery demand alone.

Our main novel contribution is to quantify the material savings from the displacement of NSBs by V2G and SLBs in general, and, in particular, the trade-off between battery reuse (material savings thanks to NSB displacement) and battery recycling (material savings thanks to higher recycled content in battery production). Previous studies could not assess these savings because they did not perform a material flow analysis that included both the demand for electric vehicle batteries and the demand for stationary storage.

We made the following changes to the manuscript to better compare our results with previous studies:

“Although V2G and SLBs can fully cover the demand for new stationary storage in later years (2034 and 2038, respectively), some NSBs will still be in use in 2050 due to their long lifetime. In an accelerated EV penetration scenario, V2G and SLBs can cover the full demand for new storage capacity about 3 years earlier compared to the baseline. These results are in line with previous studies, which estimated that EV batteries could fully cover the demand for stationary storage starting in 2030 (Xu et al., 2023)¹⁵, and that SLBs could cover the stationary storage needed to support photovoltaic production in France and Germany from 2036 onward (Kastanaki and Giannis, 2023)¹⁶.”

“We find that battery reuse reduces primary and peak primary material demand even though it reduces recycled content, *i.e.*, the share of recycled materials in newly produced batteries, in the short term but reaches similar levels as the no reuse scenario once battery demand stabilizes. This finding contradicts previous studies, which claimed that reuse would increase primary material demand because they ignored the displacement of NSBs by SLBs; and reduce the recycled content of LIBs

because they implicitly assumed an infinite demand for SLBs whereas we limit the amount of batteries that are reused by the demand for stationary storage^{15,19,20,26}.”

“Exploring all scenario configurations and aggregating over all materials, we find that the cumulative material savings from recycling range between 9-18% (8-15% for Li) for hydrometallurgical recycling efficiencies and 17-32% (18-33% for Li) for direct recycling efficiencies. Xu et al. (2020) found this value to be 20-23% for lithium with a recycling efficiency of 95%,¹⁸ which is close to our direct recycling scenario. The higher range we observe can be attributed to several factors including a faster electrification of the (European) vehicle fleet compared to their global study, making more batteries available for recycling sooner; and a slower growth rate of the vehicle fleet, which reduces the gap between the volume of batteries that become available for recycling and the demand for new batteries. Our lower end estimations are due to the consideration of less efficient recycling technologies. Some materials, e.g., phosphorus, are recovered at low efficiencies even in direct recycling scenarios.”

“Regardless of battery reuse, the recycled content ranges from 25% to 45% by 2050 depending on the scenarios considered for EV and V2G adoption (see SI 3 for a breakdown per battery material). For lithium specifically, the recycled content ranges from 0.6-5% for hydrometallurgical recycling and 1-10% for direct recycling. This value overlaps with previous findings of 5.2-6.2% by Kastanaki and Giannis (2023).¹⁶ We attribute our wider range to the larger solution space we explored by including more parameters such as stationary batteries and vehicle-to-grid.”

15. Xu, C. *et al.* Electric vehicle batteries alone could satisfy short-term grid storage demand by as early as 2030. *Nat Commun* **14**, 119 (2023).

16. Kastanaki, E. & Giannis, A. Dynamic estimation of end-of-life electric vehicle batteries in the EU-27 considering reuse, remanufacturing and recycling options. *Journal of Cleaner Production* **393**, 136349 (2023).

18. Xu, C. *et al.* Future material demand for automotive lithium-based batteries. *Commun Mater* **1**, 99 (2020).

19. Dunn, J., Slattery, M., Kendall, A., Ambrose, H. & Shen, S. Circularity of lithium-ion battery materials in electric vehicles. *Environ. Sci. Technol.* **55**, 5189–5198 (2021).

20. Bobba, S. *et al.* Bridging tools to better understand environmental performances and raw materials supply of traction batteries in the future EU fleet. *Energies* **13**, 2513 (2020).

26. Aguilar Lopez, F., Billy, R. G. & Müller, D. B. Evaluating strategies for managing resource use in lithium-ion batteries for electric vehicles using the global MATILDA model. *RCR* **193**, 106951 (2023).

6. In the context of V2G, fluctuation from renewable energy becomes very important influence to the system. The interesting point is that, the fluctuation conditions in each country in the European cases are significantly different due to the ratio of renewable energy to demand, grid scale, and etc. You need also to discuss this issue to measure the applicability of V2G in different countries in EU.

Thank you for pointing this out. We added the following paragraph to the end of the discussion section to address this limitation:

“Throughout this study, we compared aggregate storage demand with aggregate storage availability without considering bottlenecks in the electricity grids that connect centers of storage demand with centers of storage supply. We thus overestimated the effective storage capacity that V2G may supply. However, since V2G has the potential to supply more than twice the anticipated demand for stationary battery storage in the long term (see Figure 1), it seems likely that V2G could fully supply the storage demand in the long term even when accounting for bottlenecks. Future work could combine our material flow analysis with spatially explicit energy system models² that compute storage needs at various points throughout Europe.”

2. Victoria, M., Zhu, K., Brown, T., Andresen, G. B. & Greiner, M. The role of storage technologies throughout the decarbonisation of the sector-coupled European energy system. *Energy Conversion and Management* **201**, (2019).

Reviewer #2: [your comments in italics]

The authors present a distribution delay forecasting model to forecast several characteristics of waste batteries from full-electric vehicles (BEV) in the EU for the coming ten years. The lifetime distribution is similar to that used in <https://doi.org/10.1038/s43246-020-00095-x>, hence clarifications on the innovative contribution beyond the latter publication have to be highlighted better, taking into account that materials are sourced in a world market. Furthermore, it is not clear why the Economic burden of recycling LIBs was excluded from the scope of the study, even though they use the same cell chemistry in BEV batteries. The forecasting model predicts the waste stream size, second use potential, recycling potential and the potential of vehicle-to-grid. Results from this study confirm those from previous studies for different regions and time horizons.

General comments:

- Lines 37-43 describe previous work. This should include some of the recent literature on EOL i.e. <https://doi.org/10.1016/j.jclepro.2023.136349>; <https://doi.org/10.1016/j.resconrec.2021.106061> etc. in the

Introduction section:

- The Introduction part should be revised to improve its logical structure.

- There is not enough background regarding why models/functions for estimating EOL LIBs are important, what has been done and found in the previous studies, and what is still lacking.

- The authors presented some previous studies from line 58, but these contents seem too separate from the above contents. No good transitions and clear logic in these parts.

- Additionally, what is the novelty of this study to differentiate it from the previous research? Therefore, the current Introduction doesn't provide clear and comprehensive descriptions of the necessity and importance of this study.

Thank you for your feedback on the introduction's contents and structure, and for referring us to these important studies. We clarified the gap this study is addressing and added additional literature, including your suggested references (Kastanaki and Giannis (2023), Shafique et al. (2023), and Xu et al. (2020)), to justify and clarify the novelty. The literature review now reads:

"Xu et al. (2023) have concluded that electric vehicle batteries can satisfy stationary battery storage demand in the EU by as early as 2030 but they did not consider the resource implications of displacing new stationary batteries (NSBs) by V2G and SLBs¹⁵. Other studies have assessed the recycling potential of EV batteries: Kastanaki and Giannis (2023) found that SLBs in Germany and France could cover 27-70% of the stationary storage needs for photovoltaic systems, and that recycled lithium could meet 5.2-6.2% of the lithium demand for EV batteries from the EU by 2030¹⁶. Similarly, Shafique et al. (2022) estimated that recycled lithium could meet 4-6% and 7-8% of the lithium demand from EV battery production by 2030 in China and the US, respectively¹⁷. In a global study, Xu et al. (2020) found that recycled lithium could reduce the cumulative lithium demand from EV battery production by 20-23% until 2050¹⁸. All these studies focused on the LIB demand for EV batteries, but their scope excluded the LIB demand for stationary storage. They thus could not assess how V2G or reuse would affect the

primary material demand from both electric mobility and stationary storage. Studies on reuse have arrived at conflicting results. While Dunn et al. (2021) have claimed that reuse reduces the circularity of battery materials because it delays their availability for recycling¹⁹, Bobba et al. (2020) suggested that reuse is an important circular economy strategy that maximizes the use of existing materials²⁰. Aguilar Lopez et al. (2022) found that used LIBs from EVs can displace some of the LIBs that would otherwise be needed to cover the demand for battery replacements²¹. However, these studies have not considered that SLBs can displace NSBs in providing stationary storage. Without these dynamics, the resource implications of implementing SLBs and V2G cannot be fully grasped. Since energy security is predicated on attaining material security, new modelling approaches are needed.”

15. Xu, C. *et al.* Electric vehicle batteries alone could satisfy short-term grid storage demand by as early as 2030. *Nat Commun* **14**, 119 (2023).
16. Kastanaki, E. & Giannis, A. Dynamic estimation of end-of-life electric vehicle batteries in the EU-27 considering reuse, remanufacturing and recycling options. *Journal of Cleaner Production* **393**, 136349 (2023).
17. Shafique, M., Rafiq, M., Azam, A. & Luo, X. Material flow analysis for end-of-life lithium-ion batteries from battery electric vehicles in the USA and China. *Resources, Conservation and Recycling* **178**, 106061 (2022).
18. Xu, C. *et al.* Future material demand for automotive lithium-based batteries. *Commun Mater* **1**, 99 (2020).
19. Dunn, J., Slattery, M., Kendall, A., Ambrose, H. & Shen, S. Circularity of lithium-ion battery materials in electric vehicles. *Environ. Sci. Technol.* **55**, 5189–5198 (2021).
20. Bobba, S. *et al.* Bridging tools to better understand environmental performances and raw materials supply of traction batteries in the future EU fleet. *Energies* **13**, 2513 (2020).
21. Aguilar Lopez, F., Billy, R. G. & Müller, D. B. A product–component framework for modeling stock dynamics and its application for electric vehicles and lithium-ion batteries. *Journal of Industrial Ecology* **26**, 1605–1615 (2022).

The Methodology part:

- Line 339: We assume that the lifetime of all vehicles follows a normal distribution with a mean of 15 years; however in reality some of the batteries have issues in their first 5 years and reach their EOL. why the authors neglected it. There are no descriptions of this method/model, so please clarify this.

Thank you for this remark. Under our assumed normal distribution, 2.5% of all vehicles reach EOL within their first 5 years. We added the following clarification to SI 1.2:

“As can be seen in Figure 2, in our model most vehicles (68%) reach their end of life when they are between 10 and 20 years old. The remaining vehicles have a longer or shorter in use time, which accounts for differences in lifestyle choices and use habits. About 2.5% of all vehicles remain in use

for less than 5 years, which accounts for accidents, battery failures, and other causes for early obsolescence.”

- Line 368, what is C in the battery materials?

Thank you for pointing this out. We use “C” for carbon as an abbreviation for “graphite” in the materials list. We have changed all references from “C” to “graphite”.

- In the methodology section, 1.8 Recycling efficiencies why C is not mentioned, as C is mentioned for battery materials?

Thank you for bringing this to our attention. The recycling efficiency for “C” is listed under “graphite”. For consistency, we replaced the abbreviation “C” by graphite throughout the manuscript (see your previous comment).

- As there were different types of materials were used in each LIBs, however, there is no clear description of materials composition of materials for batteries in this section, this should be elaborated.

Thank you for raising this issue. We recognize that different types of LIBs require different materials. In the main manuscript, we aggregate over all materials to reduce the sensitivity of our results to changes in battery chemistry assumptions. In the supplementary information, we detail our battery chemistry assumptions in section 1.7., and we show the disaggregated material demand for each element in section 3. We describe this procedure in the methodology section.

“For the split of battery chemistries in new EVs, we follow a baseline scenario defined by Bloomberg New Energy Finance⁵⁰ (SI 1.7). When calculating the raw material needs for battery production, we aggregate over the battery materials (Li, Co, Ni, P, Mn, graphite) to reduce the sensitivity of our results to the individual materials contained in future battery chemistries. In the SI, we report the material needs for each element (SI 3).”

50. Bloomberg New Energy Finance. *Electric Vehicle Outlook 2021*. <https://about.bnef.com/electric-vehicle-outlook/> (2021).

- How it was assumed for vehicle-to-grid for each battery and vehicle type? it should be demonstrated.

Thank you for raising this issue. The assumptions underlying the V2G participation can be found in SI 1.6, where we have added the following statement to clarify your concerns regarding vehicle and battery types:

“We assume the same V2G penetration rate for BEVs of all sizes regardless of their battery chemistry.”

The Results and discussion part:

- Fig.3 should be revised to make it better.

Thank you for your suggestion. In accordance with the improvements suggested by Reviewer 3, we have added the recycled content to the figure legend in Figures 2 and 3.

- *The descriptions of the results should be made clear to indicate which scenario is the data/results for.*

Thank you for pointing this out. We have made several edits throughout the result section to better describe what scenarios the results are for.

- *The article considers the second use of lithium-ion batteries scrapped from electric vehicles and sets different utilization scenarios. What is the lifetime of the second use of a battery? How is it determined?*

Thank you for pointing this out. We added the following clarification to the SI 1.3:

“Given the lack of second-life batteries with current EV battery chemistries, there is little evidence on which to base assumptions about the lifetime of future second-life batteries²⁰. Here, we assume a lifetime of 6 years with a standard deviation of 2 years for second-life batteries in accordance with the scant literature on the topic¹⁸.”

18. Casals, L. C., Amante García, B. & Canal, C. Second life batteries lifespan: Rest of useful life and environmental analysis. *Journal of Environmental Management* **232**, 354–363 (2019).

20. Martínez-Laserna, E. *et al.* Battery second life: Hype, hope or reality? A critical review of the state of the art. *Renewable and Sustainable Energy Reviews* **93**, 701–718 (2018).

In the methodology section of the main manuscript, we explain that:

“All SLBs are assumed to remain in stationary applications until their storage capacity degrades to 60% of the initial storage capacity, which is assumed to happen within 6 years with a standard deviation of 2 years, in accordance with the scant literature on the topic⁴⁵ (SI 1.3).”

45. Casals, L. C., Amante García, B. & Canal, C. Second life batteries lifespan: Rest of useful life and environmental analysis. *Journal of Environmental Management* **232**, 354–363 (2019).

The whole manuscript should be checked very carefully since there are many inconsistent terms, error notations and expressions, and even a lack of important data sources. All of these make it difficult to read and weaken the quality of this paper.

Thank you again for your careful reading of our manuscript. Besides implementing your suggestions and the suggestions from reviewers 1 and 3, we have checked the entire manuscript and the supplementary material for any remaining inconsistencies.

Reviewer #3: [your comments in italics]

Dear authors and editor,

I have reviewed the manuscript and supporting information drafts entitled “On the potential of vehicle-to-grid and second-life batteries to provide energy and material security”, and I find the research work to be of high quality, well-reported and transparent. The topic area is of high relevance, and the paper provides new knowledge, especially in terms of the material demand impacts of using vehicle batteries for vehicle-to-grid services. Other than a few minor corrections, my recommendation is that this submission should be accepted for publication.

Proposed minor revisions:

Main manuscript text row 59 of main manuscript: The abbreviation “NSB” has not yet been introduced, this is done later in the paragraph, on row 62. Please swap this between the first and second instance.

Thank you for pointing this out. We now introduce the acronym after the first instance.

Main manuscript Figure 2: The red line representing the projected recycled content share is not included in the legend. Differing from the other line colors, the meaning of the red line can be figured out from the figure itself, but for readability and clarity, it is recommended to include the red line meaning in the legend.

Thank you, we have added the recycled content to the legend of Figures 2 and 3.

Main manuscript Table 1, Row Costs, Column SLB: It is proposed to reformulate “used” to “reused”.

Thank you, we implemented your suggestion.

Main manuscript text row 236-238: It is claimed that a benefit of delaying large-scale recycling of EV batteries is that more time becomes available for improving recycling procedures. I question this argument and find it superficial. It can equally well be argued that learning typically relates to accumulated throughput, and without large volumes of batteries coming to recycling, there will be less learning and less improvement, i.e., calendar time is of less importance. My recommendation to the authors is to revisit this specific point and revise the argument.

Thank you for pointing this out. We shifted that sentence to the end of the paragraph and modified it to read:

“Finally, by delaying the large-scale recycling of EV batteries, reuse on the one hand provides industry with more time to increase both its recycling efficiency and capacity but on the other hand limits the amount of scrap material available for refining recycling processes.”

Main manuscript text row 236-238: It is recommended to remove a part of the last sentence in the paragraph, i.e. rephrase “which would contribute to a more sustainable future with increased energy and material security.” to “which would contribute to increased energy and material security.” The suggested contribution to a “sustainable future” is vague and speculative, lowering the credibility of the remaining claim.

Thank you for pointing this out. The sentence now reads as:

“Car bans in city centers, increased investment in public transportation and bike infrastructure, and smart city design could reduce both the vehicle fleet and overall energy consumption, which would contribute to increased energy and material security.”

SI text row 73: The heading not connected to adjacent text. Please shift the heading to the following page.

Thank you, we implemented your suggestion.

REVIEWER COMMENTS

Reviewer #1 (Remarks to the Author):

The authors have responded well and revised the manuscript sufficiently.

Reviewer #3 (Remarks to the Author):

I have reviewed the manuscript and supporting information drafts entitled “On the potential of vehicle-to-grid and second-life batteries to provide energy and material security”, and I find the research work to be of high quality, well-reported and transparent. The topic area is of high relevance, and the paper provides new knowledge, especially in terms of the material demand impacts of using vehicle batteries for vehicle-to-grid services.

After my re-review of the revisions made, and responses given to all reviewers' questions, my recommendation is that this revised submission should be accepted for publication.

Reviewer #4 (Remarks to the Author):

While I find the study interesting, there are numerous issues that need to be addressed.

The introduction effectively highlights the novelty and contribution of this work. However, although the authors claim that this work could assess the resource implications of reuse and recycling by considering the competition between NSBs, V2G, and SLBs, it is not elaborated sufficiently in the results. In addition to the existing results, the authors should explain the implications of reuse on the recycling.

Assumptions about the EV lifetime are simplistic and lack of references. At the very least, vehicle and battery lifetimes should be considered separately. Besides, the modeling on battery degradation should be more detailed. In fact, although the battery will experience less SOC offset and energy throughput when providing frequency regulation service, it will generate a large amount of energy throughput when providing energy services like peak shaving, which will have an impact on the battery cycle life. Therefore, the authors should comprehensively survey existing references on battery degradation and

reconsider the model used in this study. In addition, the lack of formulaic expressions for the models used presents difficulties for readers to understand the methodology of the article.

The authors do not provide a clear description of battery material composition. A breakdown of material contents for each battery chemistry type is essential, which is the key input of this work. Besides, the aggregation results seems to be only a change in presentation without inherently reducing the sensitivity of the results.

Assumptions for the second-use lifetime of LIBs should be improved. The authors should explain why assume 6 years for SLBs with 80% SOH and 20 years for new batteries with 100% SOH. Such assumption seems not to be consistent with the assumptions of recycling until 60% SOH and linear degradation in this work. In addition, different chemistries have different lifetimes, which is quite important for the results. The authors should reconsider the battery degradation modelling.

Response letter

Reviewer #4: [your comments in italics]

While I find the study interesting, there are numerous issues that need to be addressed. Thank you for your comments.

1. The introduction effectively highlights the novelty and contribution of this work. However, although the authors claim that this work could assess the resource implications of reuse and recycling by considering the competition between NSBs, V2G, and SLBs, it is not elaborated sufficiently in the results. In addition to the existing results, the authors should explain the implications of reuse on the recycling.

We did explain the implications of reuse on recycling in the following paragraphs. For better visibility, we highlight the implications in bold. We feel that adding more explanation would be redundant.

We find that battery reuse reduces primary and peak primary material demand even though it **reduces recycled content**, *i.e.*, the share of recycled materials in newly produced batteries, in the short term but reaches similar levels as the no reuse scenario once battery demand stabilizes. This finding contradicts previous studies, which claimed that reuse would increase primary material demand because they ignored the displacement of NSBs by SLBs; and reduce the recycled content of LIBs because they implicitly assumed an infinite demand for SLBs whereas we limit the amount of batteries that are reused by the demand for stationary storage¹⁻⁴. However, **the in-use time extension caused by reuse also reduces the availability of recycled battery materials**, which increases the primary material demand per new battery. Under current hydrometallurgical recycling, where some metals are recovered efficiently but aluminium, graphite, phosphorus, and lithium are mostly lost (see SI 1.8), **the displacement of NSBs outweighs the higher primary material intensity of battery production** (compare Figures 2d with 2f and 2j with 2l). If there were no losses in the recycling process, then every old battery would result in a new battery. Since batteries lose some of their energy storage capability as they age (see SI 1.2 and 1.3), it would thus be more resource-efficient to recycle EV batteries directly after their automotive life to reduce primary material needs. It follows that the **primary material savings from reuse decrease as battery recycling becomes more efficient**. At direct recycling efficiencies (~90% efficiency for all materials, see SI 1.8), reuse is counter-productive purely from a resource use perspective (see supplementary Figure 14). However, potential reductions in infrastructure needs, energy consumption, and greenhouse gas emissions may still justify reuse.

Battery reuse reduces the recycled content, *i.e.*, the share of recycled materials from battery scrap in new batteries, **during the growth phase in storage demand between 2020 and 2040. Regardless of**

battery reuse, the recycled content ranges from 25% to 45% by 2050 depending on the scenarios considered for EV and V2G adoption (see SI 3 for a breakdown per battery material). For lithium specifically, the recycled content ranges from 0.6-5% for hydrometallurgical recycling and 1-10% for direct recycling. **This value overlaps with previous findings of 5.2-6.2% by Kastanaki and Giannis (2023). We attribute our wider range to the larger solution space we explored by including more parameters such as stationary batteries and vehicle-to-grid.**

The EU is in the process of mandating recycling efficiencies, which correspond to current hydrometallurgical recycling, where graphite, phosphorus, manganese, and aluminum are not recovered, and only 35-70% of lithium is recovered⁵. By considering that SLBs are only installed if they displace NSBs, we challenge the prevalent belief that battery reuse increases primary material demand². **Under the EU recycling mandate, we find that the losses in the recycling process are substantial enough that reuse, followed by recycling after the end of reuse life, is a more resource efficient strategy than recycling all EV batteries at the end of their automotive life to produce new batteries. If the materials mentioned above were recovered efficiently, e.g., with direct recycling (around 90% efficiency for all materials), the losses would become marginal, and reuse would become less attractive compared to using the recovered materials to produce (new, non-degraded) NSBs for the grid, when solely considering resource efficiency (see Figure 14 in the SI).** However, battery recycling and production will still require additional infrastructure and energy consumption, and reducing those needs may still justify reuse. Further research is needed to investigate these aspects. Overall, given the expected recycling rates for the short- to mid-term, reuse will likely increase material security by reducing primary material demand.

Reuse has several other benefits not directly analyzed in this study. **The creation of local markets for SLBs can help avoid end-of-life battery exports and increase the retainment of secondary battery materials in the EU.** Beyond stationary storage for the electricity grid, any excess capacity of SLBs could serve to electrify sectors that cannot afford other forms of stationary storage, such as remote off-grid areas, and provide uninterrupted power supply to critical infrastructures such as hospitals and water distribution systems. **Finally, by delaying the large-scale recycling of EV batteries, reuse one the one hand provides industry with more time to increase both its recycling efficiency and capacity but on the other hand limits the amount of scrap material available for refining recycling processes.**

The proposed EU regulation to mandate a minimum recycled content in batteries from 2026 onward⁵ would create a disincentive to reuse batteries since large volumes of battery scrap will be

needed to meet the recycled content mandate. As the European EV fleet has been and is set to continue growing rapidly, there is a risk that there will not be enough scrap available to meet the mandate until the EV stock stabilizes (see Figures 2 and 3), especially if retired batteries are reused. This might present an incentive for shorter battery lifetimes to maximize scrap availability to meet the requirements. The proposed regulation in its current form could thus paradoxically increase primary material demand. On the positive side, a recycling mandate may provide certainty about future developments of the battery industry and spur investments in recycling. It also presents a first step towards material-specific recycling targets, which can be crucial to ensure the recycling of materials whose recovery may not currently be economical but on which the EU is highly import-dependent. We note, however, that the mandate mainly targets materials such as nickel, cobalt, and copper, which are already efficiently recovered, but does not require the recycling of materials such as manganese, phosphorus, aluminum, graphite, and silicon, whose recovery would allow for recycling to become even more resource efficient than reuse in the long term.

2. Assumptions about the EV lifetime are simplistic and lack of references. At the very least, vehicle and battery lifetimes should be considered separately. Besides, the modeling on battery degradation should be more detailed. In fact, although the battery will experience less SOC offset and energy throughput when providing frequency regulation service, it will generate a large amount of energy throughput when providing energy services like peak shaving, which will have an impact on the battery cycle life. Therefore, the authors should comprehensively survey existing references on battery degradation and reconsider the model used in this study.

Thank you for your comment. We added more details on peak shaving and the limitations of our assumptions in Section 1.2 in the SI.

We assume that the lifetime of all vehicles follows a normal distribution with a mean of 15 years and a standard deviation of 5 years, which corresponds to the current lifetime of internal combustion engine vehicles (ICEVs) in the EU⁶. Initially, EVs were thought to have a shorter lifetime than ICEVs due to battery degradation but recent experience suggests these concerns were exaggerated^{4,7,8}. In addition, we neglect the impact of vehicle-to-grid on battery lifetime. This may seem counterintuitive because V2G leads to an increased energy throughput, which is generally known to accelerate battery ageing⁹. However, V2G also leads to slower charging rates and a lower average state-of-charge, which improve battery health and may more than compensate for the increased energy throughput^{10,11}. In fact, when traditional EVs are connected to charging stations, they are often charged as fast as possible. With V2G, the charging rate is modulated by the storage needs of the electricity grid. If the grid needs power at a particular moment, then EVs will charge below their desired charging rates or even discharge power back into the grid. If there is too much power in the grid, then EVs will charge

above their desired charging rates. For EVs to be able to adjust their charging rates in this manner, the desired charging rates must be set below the maximum capacity of the vehicle charger and hence below the rates of traditional charging. On average, EVs may be providing and consuming similar amounts of power to and from the grid for V2G but they often do not know in advance when they will be called upon to provide power and when they will be called upon to consume power. It is thus usually optimal to maintain a state-of-charge well below 100% to maximize the amount of storage power that can be sold to the grid. For frequency regulation, which is often regarded as one of the most profitable applications of vehicle-to-grid, it is optimal to maintain a state-of-charge of about 50%, for example¹². Finally, the amount of storage made available to the grid is often chosen in such a way that the battery experiences only modest deviations in its state-of-charge^{13,14}. Peak shaving, which consists of charging when electricity consumption is low and discharging when electricity consumption is high, is another application that may lead to negligible or even negative additional battery degradation^{15,16}. However, the long-term battery degradation induced by vehicle-to-grid has not been tested extensively yet. It is thus conceivable that there may be applications for which vehicle-to-grid will shorten battery life. In this study, we assumed that vehicle-to-grid does not shorten battery life given the lack of evidence for the impact of V2G on battery degradation.

We assume that EV batteries have around 80% of their initial capacity left when they reach the end of their automotive life^{17,18}. We consider that batteries older than 15 years degrade faster than younger batteries and thus consider a concave piecewise linear degradation curve. As can be seen in Figure 2, in our model most vehicles (68%) reach their end of life when they are between 10 and 20 years old. The remaining vehicles have a longer or shorter in use time, which accounts for differences in lifestyle choices and use habits. About 2.5% of all vehicles remain in use for less than 5 years, which accounts for accidents, battery failures, and other causes for early obsolescence. While it is true that batteries and EVs could have different lifetimes that result in different causes for obsolescence, we calculate the system dynamics using only one distribution to reduce the sensitivity of the model to specific assumptions and maximize interpretability.

3. In addition, the lack of formulaic expressions for the models used presents difficulties for readers to understand the methodology of the article.

Thank you for your comment. We now provide a more detailed explanation of our methodology in a new section (1.10) in the SI and refer to it at the end of the methodology section in the main manuscript. In addition, all the mathematical formulation can be found in the source code, which is made fully available with an open license to ensure reproducibility. The new section reads as:

1.10 Model description

The unconstrained model works using the parameters described above in a traditional material flow analysis way^{19–21} based on the mass balance principle. Consider for example the vehicle stock (Process 3 in Figure 4 in the main manuscript). We compute the size of the vehicle stock v_t at the end of any given year t in our planning horizon as the product of two exogenous parameters: the population and the number of vehicles per capita. Next, we will determine the number of vehicles v_t^+ that enter the fleet in year t and the number of vehicles v_t^- that are retired from the fleet in year t . The discrete lifetime distribution in Figure 2 specifies the fraction $L_{t\tau} = v_\tau^- / v_t^+$. Given the evolution of the vehicle fleet and the lifetime distribution, we can thus compute the inflows as $\mathbf{v}^+ = \Delta\mathbf{S} (\mathbf{I} - \mathbf{L})^{-1}$ and the outflows as $\mathbf{v}^- = \mathbf{v}^+ - \Delta\mathbf{S}$, where \mathbf{v}^+ and \mathbf{v}^- are row vectors of the in- and outflows in all years, respectively, $\Delta\mathbf{S}$ is a row vector of stock changes given by the first difference of the vehicle fleet, \mathbf{I} is the identity matrix, and \mathbf{L} is a lifetime matrix constructed such that $L_{t\tau}$ is in row τ and column t . In this model, all EVs that can be equipped with V2G will be equipped with V2G and all EV batteries that can be reused will be reused, regardless of whether there is or will be a demand for additional storage for the electricity grid.

In practice, the installed capacity for grid services should not exceed the demand as there may be no market for it. We thus performed a demand-constrained analysis to investigate the emerging competition between V2G, SLBs, and NSBs in satisfying the need for stationary storage. We assumed in particular that the newly installed storage capacity in any year exactly matches the demand for new capacity in that year. We prioritized the installation of the different storage technologies as follows:

- 1) The available V2G capacity is installed.
- 2) If the storage demand still exceeds the installed capacity, the available SLBs are installed.
- 3) If there is still a gap between the installed capacity and demand, this gap is exactly matched by installing NSBs.

If the available capacity of V2G or SLB storage surpasses the demand for new storage, only a fraction of that capacity is installed. The surplus of potentially available storage is not installed. Any excess electric vehicles are not equipped with V2G and any excess SLBs are directly collected for recycling.

All details about the numerical implementation can be found in the code, which is made fully available with an open license to ensure reproducibility.

4. The authors do not provide a clear description of battery material composition. A breakdown of material contents for each battery chemistry type is essential, which is the key input of this work. Besides, the aggregation results seems to be only a change in presentation without inherently reducing the sensitivity of the results.

Thank you for your comment. We attached the table *material_contents_overview.xlsx* to this submission, which provides the mass of lithium, graphite, phosphorus, nickel, manganese, and cobalt for all battery sizes and all battery chemistries. After acceptance, this table will be included in the zenodo container that will contain the code and data used in this study. The zenodo container will be freely accessible to the general public.

5. *Assumptions for the second-use lifetime of LIBs should be improved. The authors should explain why assume 6 years for SLBs with 80% SOH and 20 years for new batteries with 100% SOH. Such assumption seems not to be consistent with the assumptions of recycling until 60% SOH and linear degradation in this work. In addition, different chemistries have different lifetimes, which is quite important for the results. The authors should reconsider the battery degradation modelling.*

Thank you for your comment. We address this concern in Section 1.3 of the SI, which now reads:

1.3 Battery reuse, lifetime, and degradation

We consider three scenarios for the reuse rate of end-of-life EV batteries: no reuse, reuse of LFP batteries, and reuse of all batteries. In the LFP- and all-reuse scenarios, we consider that end-of-life EV batteries enter the second-life stock with 80% of their initial capacity. Given the lack of second-life batteries with current EV battery chemistries, there is little evidence on which to base assumptions about the lifetime of future second-life batteries²². Here, we assume a lifetime of 6 years with a standard deviation of 2 years in accordance with the scant literature on the topic¹⁷. For new batteries, we assume a mean lifetime of 20 years with a standard deviation of 4 years. We consider linear degradation curves for both second-life batteries and new stationary batteries. **Thus, the total lifetime of the batteries remains at 20 to 21 years on average regardless of their use. However, to account for the higher variability in user-specific battery degradation, EV batteries can leave the stock significantly earlier than stationary batteries and not be reused, resulting in a larger standard deviation for their obsolescence.**

References

1. Bobba, S. *et al.* Bridging tools to better understand environmental performances and raw materials supply of traction batteries in the future EU fleet. *Energies* **13**, 2513 (2020).
2. Dunn, J., Slattery, M., Kendall, A., Ambrose, H. & Shen, S. Circularity of lithium-ion battery materials in electric vehicles. *Environ. Sci. Technol.* **55**, 5189–5198 (2021).
3. Xu, C. *et al.* Electric vehicle batteries alone could satisfy short-term grid storage demand by as early as 2030. *Nat Commun* **14**, 119 (2023).

4. Aguilar Lopez, F., Billy, R. G. & Müller, D. B. Evaluating strategies for managing resource use in lithium-ion batteries for electric vehicles using the global MATILDA model. *RCR* **193**, 106951 (2023).
5. European Commission. *Annexes to the Proposal for a Regulation of the European Parliament and of the Council Concerning Batteries and Waste Batteries, Repealing Directive 2006/66/EC and Amending Regulation (EU) No 2019/1020*. (2020).
6. Oguchi, M. & Fuse, M. Regional and longitudinal estimation of product lifespan distribution: A case study for automobiles and a simplified estimation method. *Environ. Sci. Technol.* **49**, 1738–1743 (2015).
7. Peters, I. M. *et al.* The role of batteries in meeting the PV terawatt challenge. *Joule* **5**, 1353–1370 (2021).
8. Abdelbaky, M., Peeters, J. R. & Dewulf, W. On the influence of second use, future battery technologies, and battery lifetime on the maximum recycled content of future electric vehicle batteries in Europe. *Waste Management* **125**, 1–9 (2021).
9. Sarre, G., Blanchard, P. & Broussely, M. Aging of lithium-ion batteries. *Journal of Power Sources* **127**, 65–71 (2004).
10. Uddin, K., Dubarry, M. & Glick, M. B. The viability of vehicle-to-grid operations from a battery technology and policy perspective. *Energy Policy* **113**, 342–347 (2018).
11. Lehtola, T. A. & Zahedi, A. Electric Vehicle Battery Cell Cycle Aging in Vehicle to Grid Operations: A Review. *IEEE J. Emerg. Sel. Topics Power Electron.* **9**, 423–437 (2021).
12. Lauinger, D. *Vehicle-to-Grid for Reliable Frequency Regulation*. (Ecole polytechnique fédérale de Lausanne, 2022).
13. Uddin, K. *et al.* On the possibility of extending the lifetime of lithium-ion batteries through optimal V2G facilitated by an integrated vehicle and smart-grid system. *Energy* **133**, 710–722 (2017).

14. Thompson, A. Economic implications of lithium ion battery degradation for vehicle-to-grid (V2X) services. *Journal of Power Sources* **396**, 691–709 (2018).
15. Bhoir, S., Caliandro, P. & Brivio, C. Impact of V2G service provision on battery life. *Journal of Energy Storage* **44**, 103178 (2021).
16. Hytönen, Eric Johannes. *Vehicle to Grid Battery Degradation Impact*. (2023).
17. Casals, L. C., Amante García, B. & Canal, C. Second life batteries lifespan: Rest of useful life and environmental analysis. *Journal of Environmental Management* **232**, 354–363 (2019).
18. Reinhardt, R., Christodoulou, I., Gassó-Domingo, S. & Amante García, B. Towards sustainable business models for electric vehicle battery second use: A critical review. *Journal of Environmental Management* **245**, 432–446 (2019).
19. Brunner, P. H. & Rechberger, H. *Practical Handbook of Material Flow Analysis*. (Lewis Publishers, Boca Raton, London, New York, Washington D.C., 2004).
20. B. Müller, D. Stock dynamics for forecasting material flows—Case study for housing in The Netherlands. *Ecological Economics* **59**, 142–156 (2006).
21. Lauinger, D., Billy, R. G., Vásquez, F. & Müller, D. B. A general framework for stock dynamics of populations and built and natural environments. *Journal of Industrial Ecology* **25**, 1136–1146 (2021).
22. Martinez-Laserna, E. *et al.* Battery second life: Hype, hope or reality? A critical review of the state of the art. *Renewable and Sustainable Energy Reviews* **93**, 701–718 (2018).

REVIEWERS' COMMENTS

Reviewer #4 (Remarks to the Author):

I have no further comments for the paper.